# Myotube Guidance: Shaping up the Musculoskeletal System

**DOI:** 10.3390/jdb12030025

**Published:** 2024-09-17

**Authors:** Aaron N. Johnson

**Affiliations:** Department of Developmental Biology, Washington University School of Medicine in St. Louis, St. Louis, MO 63110, USA; anjohnson@wustl.edu

**Keywords:** skeletal muscle, myotube guidance, muscle shape, morphogenetic gene regulatory network, *Drosophila*, myogenesis

## Abstract

Myofibers are highly specialized contractile cells of skeletal muscles, and dysregulation of myofiber morphogenesis is emerging as a contributing cause of myopathies and structural birth defects. Myotubes are the myofiber precursors and undergo a dramatic morphological transition into long bipolar myofibers that are attached to tendons on two ends. Similar to axon growth cones, myotube leading edges navigate toward target cells and form cell–cell connections. The process of myotube guidance connects myotubes with the correct tendons, orients myofiber morphology with the overall body plan, and generates a functional musculoskeletal system. Navigational signaling, addition of mass and volume, and identification of target cells are common events in myotube guidance and axon guidance, but surprisingly, the mechanisms regulating these events are not completely overlapping in myotubes and axons. This review summarizes the strategies that have evolved to direct myotube leading edges to predetermined tendon cells and highlights key differences between myotube guidance and axon guidance. The association of myotube guidance pathways with developmental disorders is also discussed.

## 1. Introduction

Cellular guidance is a cytoskeleton-dependent morphogenetic process during which a post-mitotic cell generates long projections that interact with or connect to other cells. Axon guidance is perhaps the most studied form of cellular guidance and provides the foundation for connecting neurons throughout the nervous system [1]. Myotubes are immature myofiber precursors which also undergo cellular guidance to connect with tendon cell precursors. Myotubes extend bipolar projections to reach tendon cells, and the leading edges of myotube projections are functionally analogous to axon growth cones: leading edges and growth cones navigate through extracellular space, respond to guidance cues, and generate mass and volume during elongation [2,3,4].

The outcomes of myotube guidance and axon guidance are functionally equivalent. At the completion of myotube guidance, myofibers will be perfectly aligned to the overall body plan and attached to the skeleton via tendons. Achieving the correct myofiber orientation is essential for muscle contractions to produce optimized, coordinated movements. Similarly, axon guidance produces the critical connections within the nervous system and between the nervous system and target tissues, including myofibers, so that electrical signals are efficiently propagated throughout the organism. Myotubes and axons must overcome similar physical obstacles during cellular guidance, and often employ similar strategies. However, myotubes and axons have also developed surprisingly diverse approaches to manage some impediments, most notably, in response to the problem of delivering large amounts of proteins to leading edges and growth cones.

The cellular and molecular mechanisms that direct vertebrate muscle morphogenesis and myotube guidance remain largely unknown. In contrast, the body wall muscles in *Drosophila* embryos, which are analogous to vertebrate skeletal muscles, have provided unique insights into the role of myotube guidance in directing muscle morphogenesis and determining the final musculoskeletal pattern. Using myotube guidance in the *Drosophila* embryo as a model, I will discuss the integration of guidance cues with cytoskeletal dynamics and consider how myotubes acquire mass and volume. I will also present evidence that guidance cues alone are not sufficient for myotubes to identify predetermined muscle-tendon attachments and will propose the hypothesis that attachment site choice is mediated by heterophilic molecular interactions. Lastly, I will discuss the contribution of dysregulated myotube guidance pathways to heritable muscle diseases.

## 2. Myogenesis, from Progenitor to Contractile Myofiber

During the early twentieth century, myoblasts were recognized as the myofiber progenitors specified in the somites of vertebrate embryos [5]. Mononucleate myoblasts migrate out of the somites during embryogenesis to populate muscle masses in the head, trunk, and limbs. Myoblasts in *Drosophila* embryos were identified after the advent of molecular labeling technologies, and the myoblasts that give rise to body wall muscles were termed “founder cells” because each founder cell gave rise to an anatomically distinct muscle [6,7]. Unlike vertebrate myoblasts that are centrally specified in somites, *Drosophila* founder cells are specified near the position at which the subsequent body wall muscle will develop.

### 2.1. Discovering the Myotube: Surprises of the Syncytium

Somites in the chick embryo are easily accessible and, as such, served as one of the first models of muscle morphogenesis. In studies aimed at quantifying somite size, Heinz Herrmann noted that after the subdivision of the dermatome, myotome, and sclerotome, the “cells of the myotome become spindle shaped and oriented in an anterior-posterior orientation” [8]. The spindle-shaped cells of the myotome were, in fact, multinucleate myotubes that acquired a regulated orientation, presumably through myotube guidance. Gabriel Kardon also observed that “myotubes are precisely oriented within the [chick] limb, and their orientation correctly predicts the fiber orientation of the future individuated muscles” [9], suggesting myotube guidance also determines the shape of limb muscles.

One of the hallmarks of skeletal muscle is that mature myofibers are multinucleate. Two hypotheses were put forward in the late 1950s that considered how myoblasts might give rise to multinucleate syncytia: mononucleate myoblasts either undergo nuclear replication without subsequent cytokinesis or mononucleate myoblasts merge with each other to generate multinucleated cells. Irwin Konigsberg, in collaboration with Heinz Herrmann and others, blocked DNA synthesis in myoblast primary cultures and found treated cultures produced “multinuclear…elongated broad ribbon-like cells” [10]. Irwin Konigsberg went on to perform clonal analysis in cultured myoblasts and found that single myoblasts physically isolated from all other cells in culture could develop into “syncytial ribbon-like myotubes” [11]. These experiments showed that the merging, or fusing, of individual myoblasts creates muscle syncitia and coined the term *myotube* to describe the ribbon-like cells undergoing the morphological transition from round myoblasts to dramatically elongated myofibers. Thus, the pioneering studies into the cellular mechanisms of myogenesis hinted at the existence of regulatory mechanisms that orient myotubes with the overall body plan.

### 2.2. Myotube Specification and Development

Heritable stable transgenic tools, including reporter genes to visualize cell identity and the UAS/Gal4 system to manipulate gene expression, provided the easily accessible body wall muscles in *Drosophila* embryos with an experimental advantage over chicken embryos which are not amenable to heritable transgenics. *Drosophila* embryos have become a foundational model to identify the cellular and molecular mechanisms that direct striated muscle development (reviewed in [12]). After gastrulation, the mesoderm in *Drosophila* embryos will subdivide into cardiac, visceral, and somatic mesoderm; striated body wall muscles are derived from the somatic mesoderm. Thirty distinct body wall muscles develop per hemisegment, and each muscle expresses a distinct set of “identity gene” transcription factors that confer a unique cell identity (Figure 1A,B) [6,7]. The diversification of body wall muscle cell types initiates when Wingless and Decapentaplegic signals establish competence domains in the somatic mesoderm [13]. Cells in each competence domain respond to receptor tyrosine kinase (RTK) signals, and cells transducing the highest levels of RTK signals will form a smaller equivalence group of myoblasts that can differentiate into a founder cell. Lateral inhibition within the equivalence group produces a single founder cell, and the remaining cells develop into fusion-competent myoblasts. A total of thirty founder cells are specified per hemisegment, and each founder cell acquires a unique identity through the combinatorial expression of different identity genes (reviewed in [12]). Founder cell identity and diversification is thus acquired through the heterogeneity of identity gene expression.

Founder cells are, in fact, highly specialized myoblasts. As such, founder cells fuse with other myoblasts to form syncytial muscle cells, but interestingly, founder cells are capable only of directional fusion with fusion-competent myoblasts [6]. Founder cells do not fuse with other founder cells. Similar to observations in chick somites and limbs, *Drosophila* founder cells break symmetry and elongate during the transition into founder myotubes [3]. Founder myotubes can be mono- or multinucleate at the onset of myotube guidance, providing evidence that multinucleation is not an absolute requirement for myotube maturation.

Although myoblast fusion was first discovered in chick embryos it is still unclear if vertebrate myoblast heterogeneity generates specialized cell populations that participate in directional fusion. Myoblast specification in vertebrates involves a common set of basic helix-loop-helix transcription factors, including Myf5, MRF4, MyoD, and Myogenin, that are collectively known as myogenic regulatory factors (MRFs, reviewed in [14]). A key distinction between insect and vertebrate myogenesis is that vertebrate myoblast differentiation depends on a common MRF gene regulatory network, whereas *Drosophila* myoblast subpopulations express unique transcription factors (identity genes) that presumably activate distinct gene regulatory networks. While some *Drosophila* identity genes are orthologous to MRFs, many myoblast subpopulations differentiate without expressing MRF orthologs. Incredibly, the gene regulatory networks that direct vertebrate and *Drosophila* myoblast differentiation ultimately activate Mef2 expression, which is a transcription factor that activates or maintains muscle structural gene expression during the myoblast transition into myotubes [15,16,17,18].

### 2.3. Myotube Guidance Perfects the Musculoskeletal Pattern

The field of muscle development has largely focused on the specification and fusion of muscle precursors. What is less appreciated is that optimized movement requires myotubes to assemble into anatomical muscles with a topography that perfectly aligns with the overall body plan. Myotube guidance is the combined cellular processes of myotube leading edge navigation and myotube leading edge targeting to the correct tendons that establish the predetermined connections between the muscular and skeletal systems (Figure 1B). Myotube guidance, therefore, finalizes the musculoskeletal pattern to complement the mobility needs of the developing organism.

The distinction between myoblasts and myotubes is largely morphological, as differentially expressed genes that distinguish myotubes from myoblasts have not been experimentally defined. Vertebrate myoblasts exit the cell cycle before transitioning to myotubes; insect founder cells and fusion-competent myoblasts are post-mitotic prior to specification. The essential features of the myotube transition are the polarization of symmetrical myoblasts followed by the extension of two leading edges in opposing directions to form bipolar myotubes (Figure 2A). The myotube leading edges extend and navigate through the extracellular environment to identify muscle attachment sites: a single myotube will be attached to two tendons before the myotube assembles the contractile machinery and completes maturation into a functional myofiber. Similar to *Drosophila*, optically clear zebrafish embryos are amenable to genetic studies of muscle morphogenesis, and myotube elongation in zebrafish is accomplished through repeated rounds of leading edge protrusion and thickening that involves the basement membrane protein laminin [19]. It remains unclear if the extracellular matrix (ECM) organizes and orients nascent myotubes for elongation or if the ECM provides traction for membrane extensions, but in both insects and vertebrates, ECM proteins generate and maintain strong cell–cell adhesion at the myotendinous junction [19,20]. One can speculate that adhesive proteins in the myotendinous junction also carry out adhesive functions during myotube elongation that supply adhesive friction for leading edge protrusion. Generating myotubes of the correct shape and size is therefore dependent on the coordination of extensive physical forces, which highlights the regulatory challenges in assembling anatomical muscles that are perfectly aligned to the body plan.

Modern transgenic approaches have made it possible to visualize *Drosophila* myogenesis in live, optically clear embryos. Tools that fluorescently label individual founder myotubes have been developed which allow the cellular processes of myotube guidance to be recorded in an unperturbed setting at single-cell resolution [3]. Individual muscles have specific names derived from the position (dorsal, longitudinal, ventral) and orientation (lateral, oblique, acute, transverse) of the muscle in the embryonic hemisegment. The Longitudinal Oblique 1 (LO1) muscle is invariably positioned in the center of the hemisegment and acquires a diagonal, oblique orientation during myogenesis (Figure 1B and Figure 2B). Live imaging the LO1 founder cell revealed that myotube leading edges extend, navigate, and reach muscle attachment sites before the LO1 myotube becomes multinucleate (Figure 2A). Embryonic myotubes in zebrafish that develop into slow myofibers also complete myotube guidance as mononucleate cells [21]. These observations strongly support the hypothesis that myotube guidance and myoblast fusion are mechanistically distinct processes.

Some founder myotubes in the *Drosophila* embryo, including Ventral Transverse 1, fuse with neighboring fusion-competent myoblasts prior to and during myotube guidance [3]. Establishing the stereotypical body wall muscle pattern in *Drosophila*, however, depends on maintaining founder myotube cell identity, which is accomplished, in part, by preventing founder myotubes from fusing with each other [22]. Although myoblast fusion in vertebrates has not been shown to be directional, “slow” myotubes in zebrafish embryos remain mononucleate as they elongate across the somite [21], providing evidence that at least some vertebrate myotubes are incapable of fusing with each other during myotube guidance. The final muscle pattern is therefore dependent on myotube leading edge navigation, target site selection, and poorly understood regulatory mechanisms that prevent myotube–myotube fusion.

### 2.4. Gene Hunts to Identify Myotube Guidance Pathways

Compared to our understanding of myoblast cell fate specification, migration, and fusion, relatively little is known about the molecular pathways that direct myotube guidance [23]. Alleles that disrupt myotube guidance have been identified serendipitously [24,25], and by forward genetic screens in the Johnson [26], Olson [27], and Dickson [28] labs designed to identify myotube guidance mutants. The most recent screen leveraged an optimized transgenic system to express a cytoplasmic fluorophore in a subset of founder myotubes [26]. Myotube shape and orientation were then visualized in live embryos to identify and classify muscle phenotypes at single-cell resolution. This approach uncovered twenty-five genes that were not previously known to regulate embryonic muscle morphogenesis, including the zinc-finger transcription factor Spalt major (*salm*) (Figure 2B), the chromatin binding protein Barren (*barr*), the Hedgehog signaling regulator Patched (*ptc*), and the serine/threonine kinase Back seat driver (*bsd*). Insights from these genetic screens have built a molecular framework to understand and investigate the pathways that direct myotube guidance, and will be discussed in detail below.

## 3. Molecular Sensors at the Leading Edge

Like neurons, myotubes respond to attractive and repulsive navigational cues. Based on cell behavior, myotube leading edges were thought to be analogous to axon growth cones [29]. Some of the earliest molecular data in support of this hypothesis were the characterization of Slit-Robo signaling during *Drosophila* myogenesis, which acts as both a chemoattractant to initiate myotube leading edge elongation and as a repulsive cue that prevents myoblasts from accumulating at the ventral midline [24,30]. During myotube guidance, Slit is expressed from muscle attachment sites (tendon cell progenitors), and its receptor Robo is expressed in myotubes. A second Slit receptor, Robo2, is expressed in tendon progenitors where it cleaves the Slit ligand to provide a short-range signal to nearby myotube leading edges [31]. Limiting Slit distribution through a cleavage-dependent mechanism could be one mechanism that establishes a gradient-like environment for leading edge chemotaxis.

A second signaling pathway that regulates myotube guidance is directed by the orphan transmembrane receptor Kon-tiki (Kon). Myotubes in *kon* mutant embryos produced excess filapodia but often failed to connect with the correct tendon [28]. The filapodial phenotype suggested Kon regulates attachment site selection and not leading edge navigation. Kon functions through the intracellular adaptor protein Grip, which may act as a scaffolding protein to cluster active Kon complexes to the myotube membrane [28]. Alternatively, Grip may regulate intracellular signaling pathways downstream of small GTPases. Since the Kon ligand remains unknown, the mechanism by which Kon directs muscle attachment site choice is still unclear.

Genetic manipulation of the Slit-Robo or Kon-Grip signaling pathways affected only a subset of developing myotubes [24,28], suggesting that additional extrinsic inputs direct myotube guidance. Transcriptomic profiling of embryonic myotubes revealed that transcripts encoding fibroblast growth factor (FGF) pathway components were enriched in nascent myotubes [3]. Null mutations in the FGF receptor *heartless* (*htl*), or in the FGF ligands *pyramus* (*pyr*) and *thisbe* (*ths*), caused dramatic myotube guidance defects. *htl* mutant myotubes that expressed exogenous Htl showed largely normal muscle morphology, providing evidence that Htl acts cell-autonomously during myotube guidance. In contrast to Slit, FGF ligands are not expressed in the tendon progenitors at the segment borders. Instead, discrete foci of Pyr and Ths expression in the ectoderm between segment borders appear to act as short-range navigational cues, and embryos that ectopically expressed Pyr throughout the ectoderm showed dramatic myotube guidance defects [3]. These FGF studies show myotube leading edges respond to guidance cues secreted from multiple positions in the ectoderm.

The position of Pyr-expressing cells, in particular, is spatially consistent with a “choice point” hypothesis in which myotube leading edges make the decision to navigate in one of two possible directions in response to transducing a localized Pyr signal [3]. Optogenetic tools have been used to generate a photo-activatable Htl transgene which could be used to manipulate FGF signaling in individual myotubes and visualize navigational changes [32].

Myotube leading edges that lacked Htl often elongated to the incorrect muscle attachment site (tendon cell progenitor) and then navigated to nearby attachment sites in a random walk-like fashion [3]. This behavior suggests that arbitrary cell–cell contact is not sufficient for a myotube leading edge to remain in contact with a muscle attachment site and provides evidence that chemotaxis is not the only mechanism that regulates myotube guidance. One possibility is that heterophilic protein–protein interactions between myotubes and tendon cell progenitors, for example, those between Kon and its ligand, provide an additional layer of robustness to ensure myotube leading edges connect with the correct predetermined attachment sites.

Although the vertebrate counterparts of the FGF, Slit-Robo, and Kon-Grip signaling axes have not been characterized in the context of myotube guidance, the signaling ligand Wnt11 orients myotubes in the somitic myotome of chick embryos [33]. Wnt11 is expressed in the neural tube and acts through the highly conserved planar cell polarity pathway (PCP) to organize developing muscle fibers. Wnt11 is the only ligand known to act as a morphogenetic cue during vertebrate myogenesis, which highlights an essential role for PCP signaling in regulating muscle morphogenesis. Surprisingly, the impact of PCP on myotube guidance in insects remains unknown.

The contribution of the FGF pathway to vertebrate muscle morphogenesis also remains unclear. However, FGF signaling directs tube morphogenesis in echinoderms [34] and heart tube looping in zebrafish [35]. FGFs may therefore be conserved regulators of tube morphogenesis, which may extend to myotube guidance.

## 4. Integration of Sensory Information with Cytoskeletal Dynamics

The cytoskeleton is the ultimate effector of myotube guidance. The actin and microtubule cytoskeletons reorganize when muscle precursors transition from round, symmetrical myoblasts to elongated, bipolar myotubes. The myotube cytoskeleton was first visualized in fixed embryos with the filamentous actin (F-actin) dye phalloidin and antibodies that recognize microtubule plus- or minus-end-associated proteins [25,36]. More recently, fluorescent transgenes that express cytoskeleton localizing proteins have been used to live image cytoskeletal dynamics during myotube guidance [2,3]. These in vivo approaches have helped to further define the intracellular pathways that regulate the myotube cytoskeleton.

### 4.1. Actin Dynamics at the Leading Edge

The actin cytoskeleton plays an essential role in myoblast fusion and myotube guidance and decoupling the functions of actin regulatory proteins during these processes in vivo has proved challenging. Alleles that disrupt actin dynamics can cause both myotube guidance and myoblast fusion phenotypes. For example, the RNA binding protein Hoi polloi (Hoip) regulates the expression of multiple cytoskeletal proteins, including Myosin heavy chain, Tropomyosin, and actin, during *Drosophila* myogenesis [27]. The *hoip^1^* allele caused dramatic guidance defects that affected a majority of myotubes. Myotubes in *hoip^1^* embryos also showed reduced numbers of myonuclei, providing evidence that myoblast fusion is compromised in *hoip^1^* embryos [36]. Actin accumulates at the leading edge during myotube guidance, whereas discrete actin foci form at the site of membrane interactions during myoblast fusion [36,37]. These distinct actin features can distinguish guidance events from fusion events, and in the case of *hoip^1^* embryos, actin levels were reduced at the myotube leading edge, indicating Hoip is indeed required for myotube guidance [36]. Visualizing actin dynamics during leading edge navigation has been advanced by the *Lifeact* fusion protein, in which an F-actin binding oligopeptide is fused to common fluorophores and specifically expressed in nascent myotubes. Live imaging *Lifeact* during myotube guidance provides a platform to quantify F-actin accumulation at the leading edge and thereby determine the role of individual pathways in myotube guidance [2,3].

Two pathways have been identified that regulate actin cytoskeletal dynamics at the myotube leading edge. The first pathway involves Hoip, which is a Mef2 target gene that regulates Tropomyosin translation [36]. Myotubes in *hoip^1^* embryos failed to express Tropomyosin protein or accumulate actin at the myotube leading edge. Surprisingly, *hoip^1^* myotubes that expressed exogenous Tropomyosin accumulated actin at the leading edge. These studies provide evidence that Hoip regulates Tropomyosin protein levels and that Tropomyosin, in turn, modulates actin accumulation at the myotube leading edge [36].

FGF signaling is a second pathway that regulates actin dynamics during myotube guidance, where the FGF receptor Htl limits F-actin assembly at the leading edge. Genetic interaction studies showed that Htl antagonizes the activity of the Rho/Rac guanine nucleotide exchange factor Pebble (Pbl) and its effector Rac1 [3]. Rho/Rac GTPases are well-known regulators of the actin cytoskeleton, and biosensors have been developed to visualize Rho/Rac activity in vivo [38]. Live imaging studies of a Rho/Rac biosensor showed that Htl restricts Rho/Rac activity in myotube leading edges [3]. The navigational cues provided by the FGF ligands Pyr and Ths, therefore, activate a Htl-Pbl-Rac1 axis that modulates actin dynamics during myotube guidance.

### 4.2. Microtubule Reorganization

Microtubule organizing centers (MTOCs) can be composed of centriole-containing centrosomes that localize the microtubule nucleating and anchoring proteins. Alternatively, non-centrosomal MTOCs use the nuclear envelope as the site of anchoring and nucleation [39]. Under differentiation conditions, cultured C2C12 myoblasts transition into elongated, multinucleate myotubes that undergo microtubule reorganization from centrosomal to non-centrosomal MTOCs [40].

*Drosophila* embryonic myotubes undergo a strikingly similar process where the Rho GTPase activating protein Tumbleweed (Tum), and the kinesin Pavaroti (Pav), reorganizes the microtubule cytoskeleton [25]. Tum and Pav are effectors of the mitotic kinase Polo during cytokinesis. Even though myotubes are post-mitotic, Polo is required for correct myotube guidance [2]. Aurora kinases activate Polo in mitotic cells, but the Aurora kinases are not expressed in maturing myotubes. Instead, the serine/threonine kinase Back seat driver (Bsd) activates Polo in post-mitotic myotubes and directs microtubule reorganization, presumably by regulating Tum and Pav [2]. The mammalian orthologues of Bsd and Polo—Vaccinia-related kinase 3 (Vrk3) and Polo-like kinase 1 (Plk1)—are also required for C2C12-derived myotubes to elongate during differentiation [2]. Plk1 is also activated by Vrk3 in mammalian cells, providing evidence that Bsd regulates a conserved intracellular signaling pathway that directs muscle morphogenesis. Thus, the transition from mitosis to cellular morphogenesis is achieved through the spatially and temporally restricted expression of the Aurora kinases and Bsd.

### 4.3. Cytoskeletal Regulators Maintain Myotube Identity

Live imaging studies revealed multinucleate founder myotubes can fuse with each other in the absence of Bsd activity. Although Bsd and Tumbleweed (Tum) function in a common pathway to regulate myotube guidance [2], Tum does not appear to regulate fusion between founder myotubes [22]. Instead, Bsd reduces expression of *kirre*, which encodes a transmembrane protein that directs myoblast fusion. Limiting Kirre expression in the myogenic mesoderm may therefore be a mechanism that prevents founder myotube fusion.

The transcription factor Jumeau (Jumu) activates Bsd and Htl expression [22], suggesting that Jumu maintains muscle identity by preventing founder myotubes from fusing with each other. In addition, by regulating the expression of both Bsd and Htl, Jumu may be a master regulator of the actin and microtubule cytoskeletons during myotube guidance. The Jumu orthologue is not known to regulate muscle morphogenesis in vertebrates, but the T-box transcription factor Tbx3 regulates myofiber alignment in mice through unknown mechanisms [41]. Although the intracellular pathways that link guidance receptors with the cytoskeletal dynamics of myotube guidance are now coming to light, it remains unclear how transcription factor networks, like those involving Jumu and Tbx3, regulate muscle shape.

### 4.4. Cytoskeletal Regulators and Axon Guidance

Do common mechanisms regulate myotube guidance and axon guidance? This question has been difficult to address, in part because the guidance cues and cytoskeletal regulators that direct myotube guidance have been largely unstudied. The discovery of Slit as a regulator of both axon and myotube guidance suggested that the chemotactic mechanisms regulating cellular guidance would be largely overlapping [24,30]. However, Kon is not expressed in neurons and likely does not play a role in axon guidance. On the other hand, Hoip, Bsd, and Tum direct myotube guidance [2,3,27], and *hoip*, *bsd*, and *tum* mutant embryos show neuronal phenotypes consistent with axon guidance defects (Figure 3) [27]. These observations provide evidence that there is considerable overlap in myotube guidance and axon guidance mechanisms, while highlighting each cell type likely developed unique pathways to address cell-type specific obstacles.

## 5. Addition of Myotube Mass and Volume

The transition from myoblast to elongated myotube requires an incredible increase in cellular mass, but the mechanism for adding myotube mass remains controversial. In axons, intercalated mass addition occurs at the growth cone and along the axon shaft [42,43]. Microtubule networks have long been known to act as conduits for transporting the necessary materials for cell growth, including lipids, proteins, cytoskeletal materials, and organelles, from the cell body to the axon, and more recently, microtubules have emerged as mechanosensitive signaling hubs that regulate growth (reviewed in [4,44]).

Myotube growth could be accomplished through myoblast fusion, with individual myoblasts contributing the necessary growth materials for the myotube to increase in length and volume. However, live imaging of the LO1 myotube guidance in *Drosophila* showed the founder myoblast grows many orders of magnitude during the transition to a founder myotube, and even elongates across the embryonic segment, without fusing to fusion-competent myoblasts (Figure 2A) [3]. The LO1 founder myotube presents an example of myotube growth in the absence of myoblast fusion.

If the myotube does not grow through myoblast fusion, how then is additional mass generated? One clue may come from myonuclear positioning in mature myofibers. Individual myonuclei will adjust or “scale” in both size and activity depending on position or local signaling [45]. During myofiber maturation, microtubule networks reposition myonuclei, which is thought to diversify nuclear function within the syncytium (reviewed in [46]). During myotube maturation and myofiber formation in *Drosophila*, myonuclei are repositioned by microtubule networks in concert with key cellular events (reviewed in [46]). Myonuclei associate with the growing myotube leading edge during LO1 myotube guidance, suggesting that myonuclear dynamics may contribute to myotube growth (Figure 2A) [3]. Lateral transverse (LT) myotubes are in close proximity to LO1 myotubes during *Drosophila* myogenesis, and after LT leading edges reach muscle attachment sites, the myonuclei cluster to the center of the myotube. The myonuclei clusters then spread and reposition to the myotube poles during myotendinous junction formation. Myonuclei reposition yet again during sarcomere assembly and become evenly dispersed throughout the myofiber, which is thought to diversify nuclear function within the syncytium. Vertebrate myotubes also show highly dynamic nuclear positioning in culture, and myonuclei positioned proximal to sites of motor neuron innervation in vivo have distinct expression profiles compared to more distal myonuclei [3,47,48]. Thus, in contrast to neurons where the microtubule cytoskeleton transports nuclear-derived growth products from the cell body to the growth cone (axon leading edge), the myotube microtubule networks move nuclei to sites of growth and cytoskeletal rearrangements, where myonuclei presumably synthesize products required for key steps in myotube maturation and myofiber formation.

## 6. Is Muscle Attachment Site Selection Predetermined?

In *Drosophila*, short-range signals from the ectoderm provide navigational cues that direct myotube leading edges toward specific muscle attachment sites on tendon precursors. Live imaging studies showed that proximity of a myotube leading edge to a muscle attachment site is not sufficient for the leading edge to actually attach at that site, which led to the hypothesis that heterophilic protein–protein interactions between myotubes and tendon cells ensure muscles are targeted to predetermined attachment sites [3]. This bipartite information system would ensure that individual cells from two distinct deterministic cell populations, which are specified in separate germ layers, locate one another with high fidelity during embryogenesis. *Drosophila* tendons show molecular diversity [49], so it is plausible that the tendons attached to embryonic body wall muscles are also functionally diverse.

Single-cell sequencing approaches revealed that there is substantial diversity among muscle precursor populations during myogenesis [50,51]. Similarly, bulk RNA sequencing of adult mammalian muscles showed that anatomical muscles have distinct transcriptomic signatures [52]. *Drosophila* founder cells express a unique combination of identity genes, which is presumably required to establish the stereotypical muscle pattern [12]. One exciting possibility is that the diverse transcriptional identities of muscle precursors predetermine the specific tendons that a myotube can target via the expression of contact-dependent cell recognition proteins. A pre-programmed contact-dependent cell recognition mechanism could align the musculoskeletal pattern with the overall body plan, while being fully adaptable as the body plan evolves.

Why, then, would a bipartite mechanism be in place to regulate myotube guidance? One possibility is that the stochasticity of chemotactic gradients can be buffered by contact-dependent cell–cell interactions. Support for this hypothesis comes from visual circuit development in *Drosophila*, where the post-synaptic targets of photoreceptor neurons must be precisely chosen to ensure that visual information is accurately transduced to the optic lobe. After axon growth cones navigate to potential sites of cell contact, heterophilic binding between the transmembrane proteins Dpr11 and Dip ensures that the axon chooses the correct interacting partner to create a circuit that supports color vision [53]. The chemoaffinity hypothesis argues that axons identify targets through interactions with discrete heterophilic interactions. It seems likely that chemoaffinity strategies extend beyond the nervous system and are employed to precisely target founder myotubes to predetermined muscle attachment sites to generate the musculoskeletal pattern with high fidelity. Identifying the molecular components of contact-dependent chemoaffinity between myotubes and tendon precursors is therefore a high research priority for understanding how muscles acquire the correct shape.

## 7. Myotube Guidance and Muscle Disease

Dysregulation of muscle morphogenesis is emerging as a contributing cause of muscle disease and structural birth defects [54,55]. DNA sequencing costs have dramatically decreased over the past twenty years, and patients with rare genetic diseases are routinely sequenced to diagnose idiopathic disorders. As such, the number of whole genome sequences with disease-causing variants is increasing exponentially. Variants can now be queried in sequence databases to identify myogenic genes that could contribute to disease phenotypes.

Tropomyosins are obligate actin-binding proteins that are encoded by four loci in humans (*TPM1*, *TPM2*, *TPM3*, and *TPM4*), with *TPM2* and *TPM3* being the predominant skeletal muscle isoforms [56]. In mature myofibers, Tropomyosin localizes to sarcomeric actin and regulates the accessibility of myosin heads to their binding sites on actin thin filaments; this process controls overall contractility [57]. Pathogenic *TPM2* variants are causative of several disorders, including nemaline myopathy (NM) and cap myopathy (CM), congenital fiber type disproportion (CFTD), and distal arthrogryposis (DA) [58,59,60,61,62,63]. *TPM2* disorders are clinically characterized by extreme muscle weakness, a high proportion of hypotrophic type 1 myofibers, facial abnormalities, limb contractures, clubfoot, and webbing at the neck, elbows, or knees [64,65,66,67,68,69,70,71,72]. Muscle weakness associated with TPM2 variants can involve the diaphragm, which may require lifelong respiratory intervention [62,63,68]. This broad spectrum of clinical phenotypes has obscured a clear understanding of the pathogenic mechanisms by which *TPM2* variants cause skeletal muscle dysfunction.

Much attention has been devoted to understanding how *TPM2* variants cause sarcomere dysfunction. Thin filaments can be reconstituted in vitro to assay myosin-driven actin motility, and *TPM2* variants have been classified based on their performance in actin motility assays [73,74,75,76,77,78,79]. While these in vitro studies show how *TPM2* variants might affect muscle contractility, the contributions of *TPM2* variants to defective musculoskeletal development are just beginning to emerge.

Tropomyosin regulates cytoskeletal activity outside of the sarcomere. For example, cell migration and metastasis are Tropomyosin-dependent processes [80,81,82]. Cytoskeletal dynamics direct muscle precursor migration [83] and myotube guidance [2,3,36], and transgenic overexpression of human *TPM2* variants in *Drosophila*, mammalian cell culture, and zebrafish disrupts muscle development and muscle function [54]. In addition, phenotypic severity of *TPM2* variants in model organism assays correlated with the severity of patient phenotypes, suggesting that in vivo disease models have the power to predict the clinical severity of *TPM2* variants [54]. These studies highlight a role for Tropomyosin in muscle development and associate myotube guidance pathways with musculoskeletal disorders.

Genes associated with human myopathies extend well beyond TPM2 and encode sarcomere components, including actin (*ACTA1*) and myosin heavy chain (*MYH2*, *MYH7*); regulators of sarcomere assembly and stability, including nebulin (*NEB*) and titin (*TTN*); and regulators of contractility, including ryanodine receptors (*RYR1*, *RYR3*). Myotube guidance is complete prior to sarcomere assembly and the onset of contractility, and myosin heavy chain is not required for myotube guidance [27]. On the other hand, tropomyosin regulates actin dynamics in non-muscle cells and myopathy-associated *TPM2* variants disrupt muscle organization, suggesting that *TPM2* disorders involve a developmental component in addition to sarcomere dysfunction [54]. Developmental studies have identified additional myopathy-associated genes with orthologues that disrupt muscle morphogenesis in fish, including *MEGF10*, which encodes a transmembrane protein involved in signal transduction, *TRIP4*, which encodes a thyroid hormone receptor-interacting protein, and *DNM2*, which encodes a GTPase involved in cytoskeletal dynamics [84,85,86]. Continued in vivo modeling of myopathy genes in flies and fish will likely reveal that defects in muscle morphogenesis contribute to disease pathology.

Orthologues of the remaining myotube guidance genes highlighted in this review have yet to be associated with muscle disease. Variants of unknown significance in orthologues of *bsd* (*VRK3*), *polo* (*PLK1*), and *jumu* (*FOXN4*) have been identified in patients with genetic disease and reported in ClinVar [87], but none of the human genes are presently known to cause disease. Orthologues of *htl* (*FGFR1*, *FGFR2*), *tum* (*RACGAP1*), and *pav* (*KIF23*) are associated with skeletal and blood disorders [88], but many variants of unknown significance in these genes remain to be characterized. Transgenic expression or CRISPR-Cas9 gene editing in *Drosophila* could be used as a powerful platform to understand if patient-associated variants in myotube guidance genes disrupt myogenesis. This approach may identify conserved, disease-relevant mechanisms that orchestrate muscle morphogenesis.

## 8. Concluding Remarks

Although some of the pathways that regulate myotube leading edge navigation and muscle attachment site selection have been characterized, identifying the complete set of extrinsic factors that guide myotube leading edge navigation and targeting decisions to create a functional musculoskeletal system continues to remain one of the biggest mysteries in muscle biology. Experimental data show that short-range guidance cues act as navigational signals and suggest that contact-dependent cell recognition programs direct myotube-tendon targeting choices to add robustness in generating a predetermined musculoskeletal pattern. What is less clear is the function of certain transcription factors in myotube guidance, such as Salm and Jumu, which direct muscle morphogenesis but do not regulate muscle cell fate decisions.

One hypothesis is that Salm and Jumu are part of a morphogenetic gene regulatory network that modulates the intracellular response to guidance cues and regulates the expression of cell–cell recognition proteins. In this model, the morphogenetic gene regulatory network would direct spatially heterogenous effector gene expression in myotubes at distinct embryonic positions. Heterogeneity among myotubes would result in differential responses to navigational signals and would produce differential expression of cell recognition proteins to establish a musculoskeletal pattern that perfectly complements the body plan. A comprehensive analysis of Salm or Jumu function in a morphogenetic gene regulatory network would provide valuable insights into the mechanisms that control tissue morphogenesis in parallel to mechanisms that control cell fate specification.

Muscle regeneration is thought to reactivate molecular programs that direct myogenesis, but the contribution of myotube guidance pathways to muscle repair has yet to be established. With the exception of FGF receptors, orthologues of the myotube guidance genes discussed in this review have not been associated with muscle regeneration. In zebrafish, acute muscle injuries are repaired by muscle stem cells that form de novo myofibers through a cellular process that largely recapitulates myotube guidance [89]. Zebrafish may hold the key to uncovering the role of myotube guidance in muscle regeneration.

Principles from the studies described here extend beyond insect myogenesis and could be used to inform our understanding of skeletal muscle topographies in vertebrates. Navigational cues and cell–cell recognition modules are common features of cellular guidance that are likely involved in vertebrate myotube guidance. In addition, disease models show that pathogenic variants identified in patients with musculoskeletal diseases cause myotube guidance defects in *Drosophila* and myotube outgrowth phenotypes in fish and mammalian cells. Skeletal muscle organoids have recently been developed, which could be used to model human myotube guidance in three dimensions [90]. The conserved myogenic mechanisms between insects and vertebrates provide evidence that a deeper understanding of myotube guidance across species is needed to fully understand congenital muscle diseases.

## Figures and Tables

**Figure 1 jdb-12-00025-f001:**
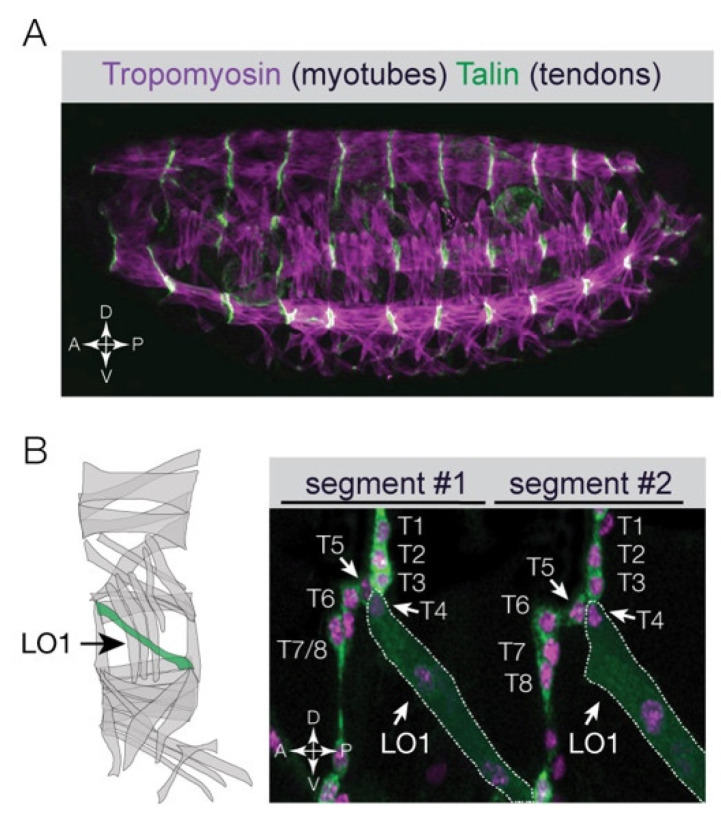
Myotube guidance ensures muscles are targeted to the correct tendon. (**A**) Confocal micrograph of a *Drosophila* embryo near the end of embryogenesis (Stage 16) labeled for Tropomyosin (myotubes, violet) and Talin (tendon cells, green). Notice the musculoskeletal pattern is precisely repeated in each hemisegment along the anterior–posterior axis. (**B**) Live Stage 16 embryo that expressed nuclear RFP and membrane-bound GFP in all tendon cells and in a subset of myotubes. Eight tendon cells (T1-T8) and one LO1 myotube were labeled (dotted white line) in two adjacent hemisegments. The LO1 myotube attached to tendon T4 in both segments despite close proximity to seven other tendon cells. Diagram shows the 30 myotubes per embryonic segment. The LO1 myotube (Longitudinal Oblique 1) is shown in green.

**Figure 2 jdb-12-00025-f002:**
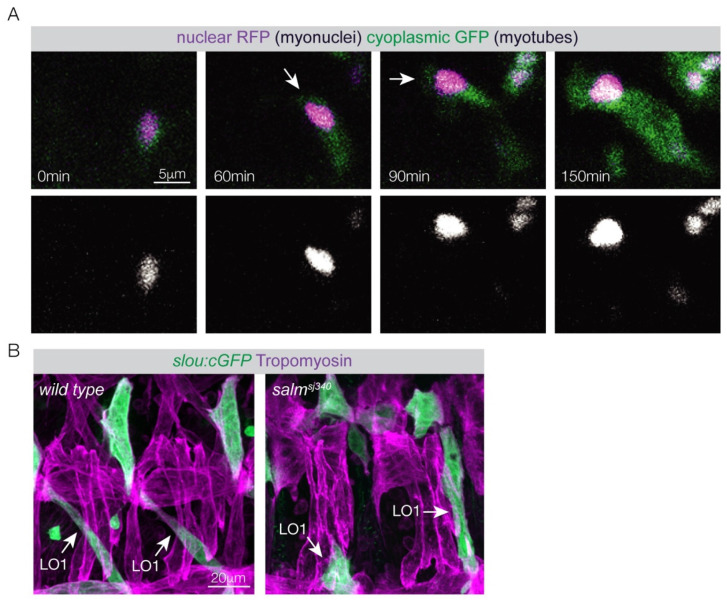
Myotube guidance. (**A**) Live imaging of an LO1 myotube that expressed cytoplasmic GFP and nuclear RFP. Arrows highlight the dorsal myotube leading edge. At 60 min the dorsal myotube leading edge reached a choice point and navigated to a muscle attachment site on the anterior of the segment. A single myonucleus followed the leading edge throughout elongation (bottom row). Notice that myoblast fusion incorporates a second nucleus at 150 min. (**B**) Confocal images of Stage 16 embryos expressing the identity gene reporter *slouch:GFP* labeled for cytoplasmic GFP (green) and Tropomyosin (violet). The pattern of *slouch:GFP* myotubes was disrupted in *salm* mutant embryos. LO1 muscles often attached to the incorrect tendons.

**Figure 3 jdb-12-00025-f003:**
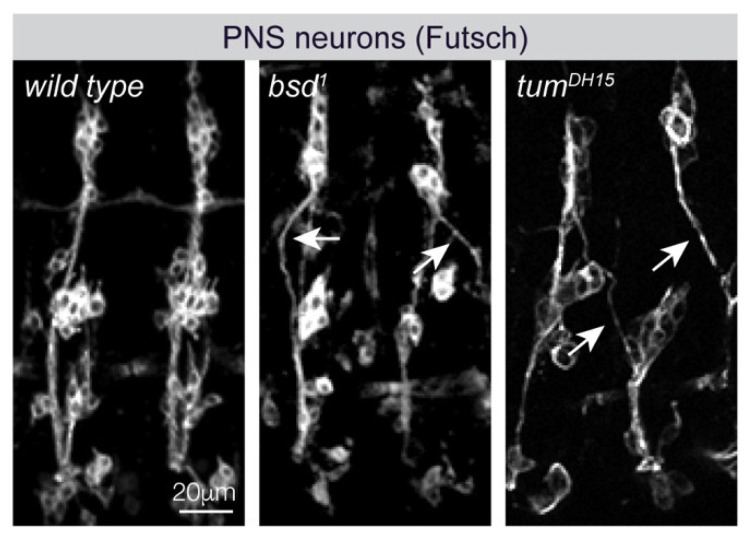
Nervous system defects in myotube guidance mutants. Confocal images of Stage 16 embryos labeled for the peripheral nervous system (PNS) protein Futsch. In wild-type embryos, multiple axons extend along a single track. Individual axons deviate from the common track (arrowheads) in *bsd* and *tum* embryos.

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
