# Peer review of "Myotube Guidance: Shaping up the Musculoskeletal System"

_jdb, 2024, doi:10.3390/jdb12030025_

Round 1
Reviewer 1 Report
Comments and Suggestions for Authors
The manuscript entitled "Myotube Guidance: Shaping Up the Musculoskeletal System” by Aaron N. Johnson is a comprehensive and well-written review. Adding the following details will further strengthen the manuscript for publication:
1. Can the review explain the positioning of myonuclei in regenerating and mature myofibers and how it migrates from central position to periphery? Does it depend on innervation?
2. Can the myogenic gene expression during myotube maturation be explained in more detail with a schematic? Are there differences between vertebrates and non-vertebrates?
3. Does basal lamina and do the ECM components play a role in myotube guidance and attachment to tendons?
Author Response
Reviewer #1
- Can the review explain the positioning of myonuclei in regenerating and mature myofibers and how it migrates from central position to periphery? Does it depend on innervation?
Response: Thank you for pointing out the need for clarification on these issues. I substantially revised the discussion of myonuclear positioning to highlight the key events and associate nuclear position with cellular functions. Here is the edit:
Here is the edit: During myotube maturation and myofiber formation in Drosophila, myonuclei are repositioned by microtubule networks in concert with key cellular events (reviewed in [44]). Myonuclei associate with the growing myotube leading edge during LO1 myotube guidance, suggesting myonuclear dynamics may contribute to myotube growth (Fig. 2A) [3]. Lateral Transverse (LT) myotubes are in close proximity to LO1 myotubes during Drosophila myogenesis, and after LT leading edges reach muscle attachment sites, the myonuclei cluster to the center of the myotube. The myonuclei clusters then spread and reposition to the myotube poles during myotendinous junction formation. Myonuclei reposition yet again during sarcomere assembly and become evenly dispersed throughout the myofiber, which is thought to diversify nuclear function within the syncytium. Vertebrate myotubes also show highly dynamic nuclear positioning in culture, and myonuclei positioned proximal to sites of motor neuron innervation in vivo have distinct expression profiles compared to more distal myonuclei [45](plus Bai 2022). Thus, in contrast to neurons where the microtubule cytoskeleton transports nuclear-derived growth products from the cell body to the growth cone (axon leading edge), the myotube microtubule networks move nuclei to sites of growth and cytoskeletal rearrangements, where myonuclei presumably synthesize products required for key steps in myotube maturation and myofiber formation.
- Can the myogenic gene expression during myotube maturation be explained in more detail with a schematic? Are there differences between vertebrates and non-vertebrates?
Response: This is a great suggestion, but a bit beyond the scope of this review.
- Does basal lamina and do the ECM components play a role in myotube guidance and attachment to tendons?
Response: Thank you for pointing out this omission. I dug into the literature and found some interesting work on the ECM and muscle morphogenesis. Here is the edit:
The distinction between myoblasts and myotubes is largely morphological, as differentially expressed genes that distinguish myotubes from myoblasts have not been experimentally defined. Vertebrate myoblasts exit the cell cycle before transitioning to myotubes; insect founder cells and fusion-competent myoblasts are post-mitotic prior to specification. The essential features of the myotube transition are the polarization of symmetrical myoblasts followed by the extension of two leading edges in opposing directions to form bipolar myotubes (Fig. 2A). The myotube leading edges extend and navigate through the extracellular environment to identify muscle attachment sites: a single myotube will be attached to two tendons before the myotube assembles the contractile machinery and completes maturation into a functional myofiber. Similar to Drosophila, optically clear zebrafish embryos are amenable to genetic studies of muscle morphogenesis, and myotube elongation in zebrafish is accomplished through repeated rounds of leading edge protrusion and thickening that involves the basement membrane protein laminin (add ref). It remains unclear if the extracellular matrix (ECM) organizes and orients nascent myotubes for elongation or if the ECM provides traction for membrane extensions, but in both insects and vertebrates ECM proteins generate and maintain strong cell-cell adhesion at the myotendinous junction. One can speculate that the adhesive proteins associated with the myotendinuous junction also play adhesive functions during myotubes elongation that supplies adhesive friction for leading edge protrusion. Generating myotubes of the correct shape and size is therefore dependent on the coordination of extensive physical forces, which highlights the regulatory challenges in assembling anatomical muscle that are perfectly aligned to the body plan.

Reviewer 2 Report
Comments and Suggestions for Authors
This is a very comprehensive review of present knowledge on the mechanisms of myotube guidance and the genes that might be involved.
The paper is well written and all the points are clearly stated.
There are some points that I think would be interesting for the potential readers and I would ask the author to add some comments on them if there is any evidence published. Many of the work, obviously, has been performed in Drosophila, zebra fish and mouse models. My queries would be:
Is there any work showing that the mechanism of muscle repair that involves satellite cell activation and fusion of new myoblasts to form new muscle after injury involves the genes that are described in the review?
Is there any work published on myotube guidance using human myoblasts?
The number of genes in which mutations lead to a congenital myopathy is very long (NEB, ACTA1, LMOD3, KLHL40, KLHL41, TNNT3, TPM2, TPM3, CFL2 ADSSL1, MYPN,TNNT1, RYR3 RYR1, SELENON, MYH7, TTN, ACTN2, MEGF10, MYH2, CCDC78, UNC45B, TRIP4 MTM1 DNM2, BIN1, SPEG, HACD1). Some of them are commented in the review. Do the author think or has evidence that some of the genes mentioned above may be also involved in myotube guidance?
Author Response
Reviewer #2
- Is there any work showing that the mechanism of muscle repair that involves satellite cell activation and fusion of new myoblasts to form new muscle after injury involves the genes that are described in the review?
Response: I checked Pubmed for muscle regeneration papers involving the genes described in the review, but only the FGF pathway emerged and these studies focused on MuSC activation. The orthologues have yet to be associated with the development of de novo myofibers during muscle repair. But the reviewer makes a great point that is now included in the Concluding Remarks. Here is the edit:
Muscle regeneration is thought to reactivate molecular programs that direct myogenesis, but the contribution of myotube guidance pathways to muscle repair has yet to be established. With the exception of FGF receptors, orthologues of the myotube guidance genes discussed in this review have not been associated with muscle regeneration. In zebrafish, acute muscle injuries are repaired by muscle stem cells that form de novo myofibers through a cellular process that largely recapitulates myotube guidance. Zebrafish may hold the key to uncovering the role of myotube guidance in muscle regeneration.
- Is there any work published on myotube guidance using human myoblasts?
Response: Great point. This is definitely a topic for future considerations, which I have added to Concluding remarks. Here is the edit:
Skeletal muscle organoids have recently been developed, which could be used to model human myotube guidance in three dimensions.
- The number of genes in which mutations lead to a congenital myopathy is very long (NEB , ACTA1, LMOD3, KLHL40, KLHL41, TNNT3, TPM2, TPM3, CFL2 ADSSL1, MYPN,TNNT1, RYR3 RYR1, SELENON, MYH7, TTN, ACTN2, MEGF10, MYH2, CCDC78, UNC45B, TRIP4 MTM1 DNM2, BIN1, SPEG, HACD1). Some of them are commented in the review. Do the author think or has evidence that some of the genes mentioned above may be also involved in myotube guidance?
Response: Thanks for the suggestion. Some of these genes may in fact be involved in myotube guidance. I have highlighted some in the text. Here is the edit:
Genes associated with human myopathies extend well beyond TPM2, and encode sarcomere components, including actin (ACTA1) and myosin heavy chain (MYH2, MYH7), regulators of sarcomere assembly and stability, including nebulin (NEB) and titin (TTN), and regulators of contractility, including ryanodine receptors (RYR1, RYR3). Myotube guidance is complete prior to sarcomere assembly and the onset of contractility, and myosin heavy chain is not required for myotube guidance (27). On the other hand, tropomyosin regulates actin dynamics in non-muscle cells and myopathy-associated TPM2 variants disrupt muscle organization, suggesting TPM2 disorders involve a developmental component in addition to sarcomere dysfunction (54). Developmental studies have identified additional myopathy associated genes with orthologues that disrupt muscle morphogenesis in fish including MEGF10, which encodes a transmembrane protein involved on signal transduction, TRIP4, which encodes a thyroid hormone receptor interacting protein, and DNM2, which encodes a GTPase involved in cytoskeletal dynamics (85-87). Continued in vivo modeling of myopathy genes in flies and fish will likely reveal that defects in muscle morphogenesis contribute to disease pathology.

Round 2
Reviewer 1 Report
Comments and Suggestions for Authors
The comments have been addressed
Reviewer 2 Report
Comments and Suggestions for Authors
The authors have addressed my queries satisfactorily